systems biology/theoretical biology/ biocomplexity

system, uncertainty, diversity, flexibility, combinability, entropy

**Author for correspondence:**
Bernard Thierry
e-mail: bernard.thierry@cnrs.fr

# Measuring complexity in organisms and organizations

Nancy Rebout[1,2], Jean-Christophe Lone[1], Arianna De Marco[2,3], Roberto Cozzolino[3], Alban Lemasson[4] and Bernard Thierry[1]

[1]Physiologie de la Reproduction et des Comportements, CNRS, INRAE, Université de Tours, Nouzilly, France
[2]Fondazione Ethoikos, Radicondoli, Italy
[3]Parco Faunistico di Piano dell'Abatino, Poggio San Lorenzo, Italy
[4]EthoS (Ethologie Animale et Humaine), Université de Rennes, Université de Normandie, CNRS, Rennes, France

 NR, 0000-0002-7071-0011; J-CL, 0000-0002-5689-1370; ADM, 0000-0002-3681-0254; RC, 0000-0002-2654-7261; AL, 0000-0001-8418-5601; BT, 0000-0002-8065-093X

While there is no consensus about the definition of complexity, it is widely accepted that the ability to produce uncertainty is the most prominent characteristic of complex systems. We introduce new metrics that purport to quantify the complexity of living organisms and social organizations based on their levels of uncertainty. We consider three major dimensions regarding complexity: diversity based on the number of system elements and the number of categories of these elements; flexibility which bears upon variations in the elements; and combinability which refers to the patterns of connection between elements. These three dimensions are quantified using Shannon's uncertainty formula, and they can be integrated to provide a tripartite complexity index. We provide a calculation example that illustrates the use of these indices for comparing the complexity of different social systems. These indices distinguish themselves by a theoretical basis grounded on the amount of uncertainty, and the requirement that several aspects of the systems be accounted for to compare their degree of complexity. We expect that these new complexity indices will encourage research programmes aiming to compare the complexity levels of systems belonging to different realms.

## 1. Introduction

Understanding complexity has become a major issue in biological and social sciences as well as in other research fields. A central question bears upon the forces that would drive biological and cultural evolution towards increasing states of complexity [1–11].

A number of questions have also been formulated with regard to the role of complexity in the evolution of living organisms and social organizations: does the stability of ecological communities depend on their complexity [12]? Do complex social systems need complex communicative signals [13]? Have enhanced cognitive abilities evolved as a response to the complexity of social life [14]? Does the complexity of human societies correlate with hierarchical organization [15] or the spread of beliefs in moralizing gods [16]? Is the gross domestic product of a country explained by its economic complexity [17]? However, progress on these issues has been slow because empirical research is hindered by the lack of a well-grounded, operational measure of complexity.

Like beauty or structure, complexity lies in part in the eye of the beholder, somewhere between order and randomness, which makes it difficult to define in an absolute sense. In dynamical systems, for example, both periodic and random processes are considered simple, while complex and chaotic processes lie in between [18]. Looking for the distinctive characteristics of complex systems, it has been proposed that they feature properties such as high dimensionality, involvement of nonlinear dynamics, occurrence of feedback loops, lack of central control or emergence of self-organization; even though these properties appear intuitively sound, as of yet, there is no agreement about them since none of them constitute a necessary condition for complex systems to arise [19–22]. The situation changes significantly, however, if we look at the outcomes of systems rather than at the nature of complexity. Given that the behaviour of complex systems is notoriously difficult to predict, it is widely acknowledged that the ability to produce uncertainty is their most prominent characteristic [21,23].

Here, we introduce new metrics that purport to quantify the degree of complexity of systems, based on the amount of uncertainty they can produce, irrespective of any assumptions regarding the nature of complexity. This should be of interest to all fields concerned with the complexity of biological organisms and social organizations.

## 2. Complexity indices

In information theory, algorithmic complexity captures the link between complexity and uncertainty in terms of compressibility: it states that the complexity of a system is equal to the size of the minimal computational resources required to generate this system [24]. It should be noted that complete randomness also corresponds to minimal uncertainty, so complexity is intermediate between a random sequence and a perfectly orderly one as mentioned above [19,25]. However, living organisms and social organizations never exhibit complete disorder.

As algorithmic complexity cannot be computed, Shannon's entropy formula is generally used instead to measure the uncertainty regarding the outcome of a random variable associated with a given probability distribution [26]. This introduces the concept of information entropy, a value quantifying the information as well as the degree of predictability of the information, which links information and complexity:

$$H = -\sum_{i=1}^{S} p_i \log p_i. \tag{2.1}$$

$H$ is Shannon's entropy (or Shannon's uncertainty) index, $S$ is the number of possible outcomes of the variable and $p_i$ is the probability of occurrence of each outcome $i$.

$H$ varies from near zero (lowest diversity when one outcome likely occurs and all other outcomes are unlikely) to a maximum value of $\log S$ (highest diversity when all outcomes are equally probable).

In a system comprising different categories of elements, the Shannon index ($H$) quantifies the unpredictability of the outcomes of a variable. Given its unifying potential, Shannon's entropy has been used in various fields and particularly in biology, where it has been applied in innumerable ways to assess the diversity of living systems. As previously mentioned, however, organisms and organizations cannot be reduced to the number and distribution of their basic constituents only. Therefore, focusing the use of Shannon's entropy metric on the diversity of system elements falls short of accounting for the whole complexity of biological and cultural systems. To reconcile the measure of uncertainty with the structure and function of these systems, the calculation of diversity should be extended to further dimensions of systems.

From the simple statement that a system is a set of elements that are interrelated [27], it follows that systems are composed of a variable number of elements, but also that the elements themselves can be variable, and that they can associate in variable patterns. This leads us to consider measures that can

reflect three major dimensions of system complexity: diversity, flexibility and combinability. Much effort has been devoted to assessing the diversity of organisms and organizations by measuring the number and variety of their building blocks [4,6,28–30]. However, diversity is only one component of complexity and we still lack indices capable of capturing the full extent of complexity. The measurement of complexity requires that all three dimensions are accounted for. As we develop below, the measures of diversity, flexibility and combinability will each be calculated using Shannon's formula, applying it to different variables in corresponding sample spaces.

## 2.1. Diversity index

When analysing a system, we have first to specify what the elements of that system are, i.e. its basic units. These may be cells in the study of organisms, or signals in the study of communication systems, for instance. According to Shannon, a system is more diverse, the more it contains a greater number of categories of elements and the more balanced the number of elements is among the categories. Shannon's metric of entropy consists of two components called richness and evenness in ecology. Richness is the number of possible outcomes, i.e. types or categories of a variable. It is a popular measure of diversity/complexity as it is relatively easy to count cell types in organisms [31], species in ecological communities [32], signals in animal communication [13,29], structures in languages [28] or cultural variants in human societies [6,33]. Evenness refers to the heterogeneity of probability of the different categories composing the richness, whether structural or functional. In other words, evenness is the distribution law of the probabilities of the different outcomes of the variable (distribution law of $p_i$). It is the interplay between evenness and richness that can be used to address the diversity of systems.

Early on, the differentiation or specialization of system elements in discrete roles has been recognized as a clue to complexity [34]. Shannon's entropy is used to measure the diversity of phenomena as diverse as ecosystems, social relationships, communication signals or neural networks (e.g. [32,35–38]). The entropy value was, however, devised to enable the comparison of indices with a different number of outcomes. It has to be adjusted to compare systems, so we will use the relative index [32,39]

$$h = \frac{H}{H_{\max}}. \tag{2.2}$$

$H_{\max}$ is the maximal value of $H$, i.e. log $S$.

To measure $h$, we need to specify the sample space of a variable of interest, i.e. the set of all possible outcomes. In the field of genetics, for example, we can consider a sample space based on the different allelic types and their frequency in a population [40]. For a variable $v$, we calculate a relative diversity index $h_{vD}$ using formula (2.2)

$$h_{vD} = \frac{H}{H_{\max}} = \frac{-\sum_{i=1}^{S_{vD}} p_i \log p_i}{\log S_{vD}}. \tag{2.3}$$

$S_{vD}$ is the number of categories of a diversity variable $v$.

Diversity depends on the different variables describing the elements of the system under consideration. To obtain the diversity index $D$ for the system on a scale of 0–1, regardless of the number of variables, we calculate the mean of the relative diversity indices of the different variables:

$$D = \sum_{v=1}^{n} \frac{1}{n} h_{vD}. \tag{2.4}$$

$n$ is the number of variables.

Note that when the number of $h_v$ values is low, a median can be calculated instead of the mean. Investigators will need to choose whether to calculate the median or the mean depending on their dataset.

## 2.2. Flexibility index

While diversity concerns differences between categories of elements, variation can also occur within categories. Contrary to systems currently envisioned by physics and engineering that are made of discrete and relatively fixed elements, living organisms and social organizations are composed of flexible elements and parts. Elements can vary; they are able to shift from one state to another,

meaning there are variations of the elements of the system under different conditions. It can be also that there is continuous variation between the elements of a system, meaning that categories intergrade and that one element can belong to different categories. Flexibility, variability or plasticity, whatever it is called, increases the unpredictability of systems, allowing them to explore possible functional abilities and respond to changing environments, which has a clear adaptive value [19,41].

To calculate a flexibility index, we have to specify the variables expressing the flexibility of system elements, and the sample space for each variable. Possible outcomes should be defined according to the logic of Shannon's entropy; for each variable of interest, the construction of a sample space requires that the distribution of outcome probabilities expresses the uncertainty of the system. Using the field of phenotypic plasticity, for example, we may illustrate the flexibility in terms of variation of elements under different conditions, choosing a sample space based on the proportion of different phenotypes in a population; some butterflies show two discrete seasonal phenotypes, dry-season and wet-season phenotypes [41], and, therefore, we can define a sample space based on both phenotypes as possible outcomes, which corresponds to possible switches from one phenotype to another. Note that we can also estimate continuous variations in a given butterfly phenotype by transforming measures like wing length and body size by cluster analysis. Specifically, we can construct categories of morphologically similar individuals considering a continuum between these categories, using, for instance, a cluster analysis based on soft assignment [42]. Entropy can be calculated because soft clustering algorithm gives for each element its probabilities of belonging to the different discrete categories. The more evenly the probabilities are distributed among the different discrete categories, the higher the degree of gradation among morphologically similar butterfly categories. It is, therefore, possible to use continuous variables and to calculate an index based on entropy, which makes it possible to apprehend the continuous nature of the system.

For the variable $v$, we calculate a relative flexibility index $h_{vF}$ using formula (2.2), then a relative flexibility index similar to the relative diversity index (2.3)

$$h_{vF} = \frac{H}{H_{\max}} = \frac{-\sum_{i=1}^{S_{vF}} p_i \log p_i}{\log S_{vF}}. \tag{2.5}$$

$S_{vF}$ is the number of categories of a flexibility variable $v$.

The flexibility index $F$ of the system is the mean of the relative flexibility indices of the different variables

$$F = \sum_{v=1}^{n} \frac{1}{n} h_{vF}. \tag{2.6}$$

## 2.3. Combinability index

System elements can interact and associate at different levels, which introduces a further degree of uncertainty in systems. The nature and amount of connections that occur at the dyadic level, i.e. within pairs of elements, represent a first source of uncertainty. In the study of animal behaviour, for instance, it has been proposed to measure the complexity of social groups from the number and strength of relationships between individuals [29,30,43–45]. Connections can also arise at the triadic level, i.e. between more than two elements. In some mammals, social competition drives several males to associate in alliance networks, which generates subgroups of varying size and stability [46,47]. More or less marked cliquishness, compartmentalization or modularity, irrespective of the designation given to it, is a general property of biological and social systems; it means that they are composed of multiple subunits that are structurally and/or functionally semi-independent [41]. In modular organization, subunits are arranged in parallel, as for cell organelles or segmented body parts. In hierarchical organization, subunits are arranged in nested levels where larger parts are composed of smaller parts, as for organisms, organs, cells, organelles and molecules.

Counting the number of connections, modular parts or nested levels are employed to estimate complexity both in biological and social sciences [4,5,11,15,43]. However, such methods based on separated counts remain limited. Even relatively simple systems such as bird songs can be highly combinatorial: groups of notes form syllables which are themselves assembled into phrases that are then grouped into songs, and these different subunits can appear in various combinations at multiple levels [48]. Instead of separately quantifying connectedness, modularity and nestedness, we may

consider complex systems as sets of subunits which can vary in their degree of dissociation and differentiation, as well as in the interactions that link the units composing them.

To measure $h$, we need to specify the different variables expressing the patterns of interaction and association between the system elements. Take, for example, the study of protein interactomes [49]. To deal with the networks of protein–protein interactions, we may choose sample spaces based on proteins or groups of proteins, then consider the distribution of connections. The more evenly distributed the connections are, the greater the uncertainty in protein–protein interactions. For each variable $v$, we calculate a relative combinability index $h_{vC}$ using formula (2.2), then a relative combinability index similar to the relative diversity index (2.3)

$$h_{vC} = \frac{H}{H_{\max}} = \frac{-\sum_{i=1}^{S_{vC}} p_i \log p_i}{\log S_{vC}}.$$

(2.7)

$S_{vC}$ is the number of categories of a combinability variable $v$.

The combinability index $C$ of the system is the mean of the relative combinability indices of the different variables

$$C = \sum_{v=1}^{n} \frac{1}{n} h_{vC}.$$

(2.8)

## 2.4. Complexity index

A complexity index $K$ of a given system can be drawn from its diversity, flexibility and combinability. Since the three dimension indices $D$, $F$ and $C$ are independently measured entropies (formulae 2.4, 2.6, 2.8), we can calculate $K$ by summing these three indices. If we have no assumptions about the relative importance of the three dimensions, we can assign equal weight to the three indices

$$K = D + F + C.$$

(2.9)

## 2.5. A calculation example

To illustrate the calculation of indices, we take an example from the comparative study of social systems in macaque monkeys. All macaques live in groups containing both adult males and adult females with offspring, but they display wide interspecific variation in their social relationships [50]. Some are characterized by strong social intolerance, meaning that they display a steep gradient of dominance coupled with conspicuous submission signals and a strong preference for kin partners. Other species show higher levels of tolerance, which correspond to moderate power asymmetries, a high propensity to regulate conflicts through affiliative behaviours, and a relatively low degree of preference for kin. It appears that strong social tolerance provides individuals with large degrees of freedom in social interactions and relations, whereas weak tolerance lends more weight to the influence of social status on individual behaviours, with presumably more predictable outcomes (see [50,51]). To illustrate how the correlation between complexity and tolerance can be evaluated in macaque social organization, we calculated the complexity indices of the social system in two species of contrasting social relationships: tolerant Tonkean macaques (*Macaca tonkeana*) and intolerant rhesus macaques (*Macaca mulatta*). It should be noted that the data needed to calculate the indices are still scarce, so the variables selected were those for which the necessary information was available.

### 2.5.1 Diversity

Diversity concerns the characteristics of the elements that make up a system and by which these elements can be described. Here, the elements are individuals, they can be described by their age and sex, as these two factors lead to social groups containing several demographic categories with different behaviours and statuses. The more balanced the proportions of the demographic categories are, the greater the uncertainty about the category of individual a given individual may interact with. Therefore, demographic categories can be used as possible outcomes to assess social diversity (see [50]). Based on field data in each species of macaque, we assigned individuals to three age-and-sex categories: adult males, adult females and immatures (electronic supplementary material). Calculating relative

diversity indices, we obtained $h_{vD} = 0.890$ and 0.979 in rhesus and Tonkean macaques, respectively (electronic supplementary material). We used a single variable to estimate the diversity of macaque social systems, so the diversity index $D$ was equal to $h_{vD}$ in each species.

### 2.5.2. Flexibility

Flexibility is about how elements can vary. In social systems, the elements are individuals, and their behaviour may vary according to social situations. We can use Shannon's entropy to quantify behavioural variations in individuals [29,52]. Comparative data are available in macaques for two kinds of social events: in the reconciliation that follows conflicts and involves different behaviour patterns (i.e. one social context, different behaviours), and in the occurrence of a specific facial expression, the bared-teeth display, which is observed in different social contexts (i.e. one behaviour, different contexts; electronic supplementary material). With regard to reconciliation, we differentiated between four categories of behaviours (body contact, vocal signal, facial expression, gesture). Behaviours that occur in many contexts indicate a higher degree of freedom regarding the expression of behaviours by individuals, i.e. flexibility. The sample space was defined by the proportions of behaviour occurrences: the larger the number of behaviours simultaneously occurring in a reconciliation, the higher the uncertainty of the social encounter. We calculated the relative flexibility indices, and obtained $h_{vF} = 0.560$ and 0.615 in rhesus and Tonkean macaques, respectively (electronic supplementary material). With regard to the bared-teeth display, the sample space was composed of five social contexts (affiliation, play, mating, submissive response to aggression, spontaneous submission) as outcomes: the larger the number of contexts of occurrence for this facial expression, the higher the uncertainty. For the relative flexibility indices, we obtained $h_{vF} = 0.394$ and 0.633 in rhesus and Tonkean macaques, respectively (electronic supplementary material). Lastly, we calculated the flexibility index $F$ in each species as the mean of the relative flexibility indices: rhesus macaques $F = 0.477$, Tonkean macaques $F = 0.624$.

### 2.5.3. Combinability

Combinability refers to the connections between the elements of a system. Since the elements of a social system are individuals, combinability concerns patterns of interactions and relationships between individuals. We estimated the uncertainty stemming from relationships between group members based on two kinds of social interactions. We used the distribution of social grooming among individuals at rest to assess the degree of subdivision of the group into subgroups (i.e. modules) as a function of kinship ties; and we used social conflicts by distinguishing between unidirectional conflicts (i.e. including a winner and a loser) and bidirectional conflicts (i.e. both opponents threaten or attack each other, without producing a clear winner) to assess the degree of uncertainty in the outcomes of the interactions. Grooming interactions can be exchanged between close kin partners or non-close kin partners. We reasoned that strong kinship ties may be used to reliably know which are the most frequent partners—corresponding to relatively closed clusters of related partners—whereas weaker ties make the partner choice less predictable, with less recognizable clusters of related partners. This led to a sample space where the less kin-biased the partner choices, the greater their uncertainty. We calculated the relative combinability indices and obtained $h_{vC} = 0.548$ and 0.983 in rhesus and Tonkean macaques, respectively (electronic supplementary material). With regard to social conflicts, the sample space was defined by the proportion of aggression displayed by each opponent in pairs of individuals, where the uncertainty was higher when both opponents displayed similar rates of aggression. For the relative combinability indices, we obtained $h_{vC} = 0.229$ and 0.881 in rhesus and Tonkean macaques, respectively (electronic supplementary material). Lastly, we calculated the combinability index $C$ in each species as the mean of relative combinability indices: rhesus macaques $C = 0.389$, Tonkean macaques $C = 0.932$.

### 2.5.4. Complexity

By summing (2.9) the three indices $D$, $F$ and $C$ measured in each species, we obtained the following values for the complexity index $K$: 1.76 for rhesus macaques and 2.54 for Tonkean macaques. This result is consistent with the hypothesis that system complexity increases with social tolerance among macaques. However, the bulk of the effect comes from the difference in the combinability index (figure 1). In addition, any conclusion on overall complexity would be affected by the relative weights that could be assigned to each of the three indices.

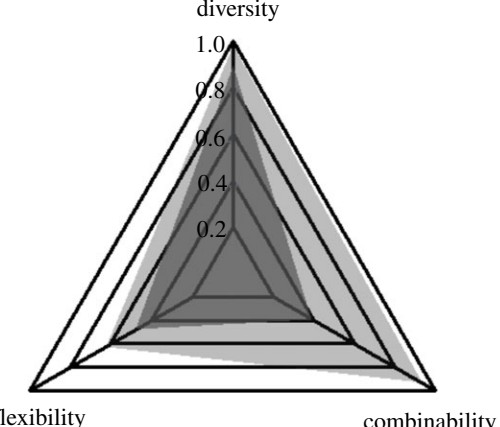

**Figure 1.** Radar plot comparing indices in rhesus (dark grey) and Tonkean macaques (light grey). Each spoke represents an index. The plot reads from the centre outward along each spoke. Scores are shown on concentric triangles beginning at 0 (centre) and increasing to 1 (outer triangle). We can see from the figure the relative contribution of each index to the discrepancy found between species regarding the complexity of their social systems.

It should be noted that we have used the example of macaque social systems to describe the calculation of the complexity index, but data from a higher number of species would be necessary to test whether the indices measured in different kinds of species are statistically different. Moreover, the calculation depends on the variables chosen, so that the collection of data on a much larger number of variables would be necessary to obtain representative indices. The variables and the corresponding sample spaces must be carefully selected, and we provide guidelines on this in the electronic supplementary material.

## 3. Discussion

Research has long focused on single features of complexity rather than acknowledging its multidimensional nature [12,22,29,53]. Diversity, flexibility and combinability each capture a part of complexity. By integrating their measurements, we may also encompass the whole complexity of biological and cultural systems. These indices based on Shannon's uncertainty differ from previous measurements by several aspects.

First, we do not make direct use of the number of parts of a system to estimate complexity. Common sense considers a large number of elements as a main characteristic of complex systems, but the number of basic components of a system is a rather crude proxy. Few people would use the number of cells to compare the complexity of a fly with those of large sponge, and it is well known that the variation in the genome size among organisms does not have simple relationships with the number of coding genes or levels of phenotypic organization [3,54]. Another example comes from testing the hypothesis that an increase in social complexity drove the evolution of enhanced cognitive abilities through the evolution of species. After decades of research, the issue has yet to be resolved, due in part to the fact that social complexity was approximated by the number of individuals per group, and cognitive performances by brain size [14,22,29].

Since Shannon's index includes the number of categories, it should be noted that it indirectly takes the number of elements into account. In the study of societies, for instance, differentiating between familiar and non-familiar partners has little relevance in small groups, while several categories of familiarity can be distinguished as the size of the groups increases. More generally, the amount of categories tends to increase with the number of elements of a system (e.g. [7,10,55]).

The inclusion of flexibility is a second distinctiveness of our proposal. To date, flexibility has been missing from works aiming to measure complexity. Yet, it is present at all levels of organisms and organizations; it conditions their adaptation, robustness and reproduction [41]. It is hypothesized, for example, that flexible social systems have evolved as a response to unpredictable environments in animals [56]. More generally, living beings are capable of learning, which adds a further layer of flexibility, the importance of which varies to a considerable extent depending upon the species. It may also be worth remembering that the ability to learn from others forms the basis upon which cultural

systems rest and which themselves are rich in changes and innovations [57]. This makes it essential to account for a significant number of flexibility variables in order to measure the complexity of systems.

Many authors have distinguished between diversity, measured by the number of categories, and modularity, measured by the number of parts. As these two kinds of variables are often considered separately, however, the paradox is that they can be one and the same thing. For instance, the number of cell types in organisms and the number of castes in insect societies may be alternatively counted as diversity (number of specialized categories) or modularity (number of specialized parts) [4,10,55]. The issue originates from the definition of the subunits making up a system. Any subunit can be decomposed into more basic subunits, and, therefore, the decision on which level to focus the analysis becomes somewhat arbitrary. By requiring that diversity and combinability be taken into account simultaneously, the tripartite complexity index obliges that a differentiation be made between the basic components of a system and their number of categories on one side, and the number of modular parts or nested levels on the other. This is a third characteristic of our proposal. Moreover, it needs to specify the basic components of the systems under consideration, thereby avoiding confusion between levels. When studying the complexity of animal communication, for instance, the flexibility of signals emitted by individuals should not be mixed with the diversity of repertoires which often applies to populations or species [37,58,59].

The measurement of combinability could be further elaborated. Our index allows for the directionality of the interaction to be taken into account, as illustrated by the second estimate of combinability based on social conflict in our calculation example. But two other components can be taken into account for combinability, namely nestedness and modularity. Nestedness means that the elements of the system are ordered in hierarchical units, which introduces an additional level of uncertainty. An index based on the proportion of the different distances in a hierarchy tree representing the links between system elements could measure this. Modularity means that system elements are ordered along parallel units. In a network, for example, the more balanced the links between the units, the greater the uncertainty in the network and, therefore, in the system; on the other hand, in a system where subgroups are clearly delineated, associations are more predictable and the network is then considered less uncertain and thus less complex. The network data can, therefore, be used to calculate an index based on the distribution of connections.

Lastly, our complexity indices go beyond a mere empirical denumbering of parts. Building upon information theory, they enable the comparison of different systems by explicitly quantifying complexity levels in terms of uncertainty. It should be emphasized that some systems may differ in some dimensions and be similar in others, so it may be useful to compare not only systems based on the tripartite complexity index, but also dimension per dimension. There is still room for improvement since each complexity dimension may be measured in multiple ways. An index is as good as the data on which it is based, so it is advisable to measure each dimension by as many variables as possible. At present, it is difficult to find in the literature the data needed to calculate the three complexity indices for living organisms and social organizations. Hopefully, the present proposal will encourage research programmes that aim to measure the variables needed to compare the complexity levels of systems in both biological and social fields.

Data accessibility. The sources of the data are provided in the electronic supplementary material.
Authors' contributions. BT and NR: conceptual design and manuscript writing; NR and JCL: mathematical formulation; ADM, RC and AL: critical comments and additions to the manuscript.
Competing interests. We declare we have no competing interests.
Funding. We received no funding for this study.

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
