## [Peer Review File · Royal Society Open Science]

Review History

RSOS-200895.R0 (Original submission)

Review form: Reviewer 1

Is the manuscript scientifically sound in its present form?

Yes

Are the interpretations and conclusions justified by the results?

No

Is the language acceptable?

Yes

Do you have any ethical concerns with this paper?

No

Have you any concerns about statistical analyses in this paper?

No

Recommendation?

Major revision is needed (please make suggestions in comments)

Comments to the Author(s)

Shannon entropy is a classic long-term concept, please at least including some empirical data from empirical systems to make the definition more solid and useful.

Review form: Reviewer 2**Is the manuscript scientifically sound in its present form?**

No

Are the interpretations and conclusions justified by the results?

Yes

Is the language acceptable?

Yes

Do you have any ethical concerns with this paper?

No

Have you any concerns about statistical analyses in this paper?

No

Recommendation?

Major revision is needed (please make suggestions in comments)

Comments to the Author(s)

In this manuscript, authors introduce three new measures to quantify the complexity of systems. Built on information theory, their measures derive from the Shannon index measure of uncertainty, uncertainty being indeed, one of the most prominent characteristics of complex systems. Introducing three different indices to quantify complexity - diversity, flexibility, and combinability -, they also better acknowledge the multidimensional nature of complexity, considered as another notable property of complex systems. Doing so, they aim to provide a more comprehensible and comparable method for the study of systems' complexity, which was indeed, so far lacking. Doing so they point out the need for common operational and comparable measures of complexity, that could be easily applied to various systems. This certainly opens new possibilities and work avenues.

However, before recommending this manuscript for publication, I feel that substantial amendments/improvements regarding its structure and content are required.

First, in my view, the state of art regarding the definition of complexity (e.g., the different approaches/methods/measures that have been used to quantify it, and the different research fields that have address it) should be further developed. Doing so, I believe the work would benefit from a clearer positioning with respect to previous research (i.e., what has been done so far to quantify and compare complexity, and to which extend and why are we failing to do so), better highlighting its added value.

In the same line, I believe that the field of research mainly addressed in this work should be more explicit. In several sections (introduction, discussion) authors seem to address the field of animal

social behavior, and in particular the study of social systems complexity (e.g., lines 48, 79). Following this, authors quantify the complexity of macaques' social system in their work example. And indeed, the study of social complexity across social systems is a hot topic. Such a contribution in the field would thus certainly be valuable.

However, the literature/framework in which this proposal is anchored, is never really explicitly mentioned, although this could substantially enrich the introduction and discussion (e.g., discussing the added value of their complexity measures with respect to the specific expectations and state of the art of the field), help the reader to identify the potential application of the work, and once more, better highlight its novelty.

E.g., introduction lines 64-68: when discussing the benefits of considering the outcomes of systems rather than their complexity per essence, it could be beneficial to the argumentation to provide a couple of examples applied to social systems to illustrate this particular point. This would also certainly ease the reading and contribute to illustrate potential applications of the proposed metrics and methodology.

It is sometimes difficult for the reader to picture at which analytic level the discussion (or calculations) is (are) made. For instance, the differences between elements, variables, types are not clear, and often confusing (e.g., lines 155-164). The use of those terms seems inter-changeable in some sections, while meaningful in others, although I'm sure their meaning might be clear to authors. I acknowledge the difficulty to find homogenous and logical terminologies to name such abstract concepts, but I think this is crucial to ensure a proper understanding of the methodology and its applications. A short section clearly defining what is meant by each 'technical' term, with a couple of examples, could be useful here.

At the end of the short introduction (line 107), could authors clearly say that the three following measures (diversity, flexibility, and combinability) are calculated using the exact same formula (Shannon's), but, applying it to different sample spaces?

Following this, the definition of the sample space appears crucial to this work. The whole method developed here thus rely on a proper understanding and definition of the sample space, for each of the three measures. However, I feel that this neither sufficiently highlighted nor explained with enough accuracy (first mentioned lines 133-136).

Another concern regards the subjectivity of the proposed metrics.

Again, if I understand correctly, the calculation of the three indices mostly relies on the specification of the sample space of each variable of interest. However, the choice of the elements/variables of interest and the definition of their sample space seems largely arbitrary to me. (Especially since the manuscript is not clearly positioned in a given field, and its associated guidelines, methodological challenges, or most prominently studied parameters not discussed). If the choice of elements/variables and the definition of their sample space is primarily determined by the study system, I'm afraid it will mainly depend on the current knowledge of the system, and the type of data available (as pointed by authors lines 280-281, 404-408). Although corrected by Hmax (lines 123-128), I can imagine that the metrics would substantially vary depending on the choice of the subunits characterizing the system (e.g., when studying social systems: population, group, individual, to interaction level, as discussed lines 373-375); the nature and numbers of elements/variables considered; and the definition of their sample space.

My question is, therefore, how representative and reliable can those indices be?

For instance, in the working example, why using a single variable for the calculation of the Diversity index? And for this variable, which seems to be a demographic variable (this also has to be clarified), why are individuals chosen, as elements and differentiated only on the basis of their

age and sex and no other characteristics e.g., sexual maturity, or considering the underlying genetic structure of the species which might be in particular relevant in species showing matriline – to which matriline cluster individuals belong -? Why not choosing a higher degree of refinement regarding age e.g., scaling age per year? Considering the full interaction between age and sex i.e., differentiating also male and female immatures? Similarly, for the flexibility index. Why quantifying behavioral variation and not relationships variation within the structure? And are the two kinds of social events considered (reconciliation and facial expression) sufficient to depict trustworthily behavioral variation across individuals' categories?

My problem here is that I feel that all those analytic choices can certainly be well justified. But they might all lead to different indices outcomes. This thus raises the question of comparability (between studies, systems). To ensure the reliable comparability of those indices, one would need to choose the exact same 'parameters' and sample space. Which, in my view represents the biggest challenge here, and therefore requires further discussion.

Could authors further discuss this, and possibly provide guidelines to help researchers how to properly choose the parameters of interest and their sample spaces, to consistently and comprehensibly investigate systems complexity?

Finally, no conclusions are drawn from the calculated indices in the working example, to answer the secondary objective of this work: "To evaluate whether complexity and tolerance correlate in macaque social organization, we calculated the complexity indices in two species of contrasting social relationships" Line 258-261. If authors do not aim to discuss their results but simply wish to detail their method, they should remove this statement. But I personally believe that concluding and discussing their results would improve the understanding of the measures and their potential applications and implications. But this remains to be done.

That being said, I am wondering whether the question isn't circular here (i.e., correlating the degree of tolerance to the calculated (social) complexity measures). Macaques' degree of tolerance is a by-product of the diversity, nature, and distribution of relationships and interactions in the groups. E.g., Compare to more tolerant systems, strong intolerance might be defined by more asymmetric and sparser relationships at the group levels, both for dominance relationships (i.e., clear dominants and subordinate status along with clear dyadic dominance relationships, and a larger proportion of dyads that likely never or very rarely directly 'encounter'), or affiliative relationships (i.e., relationships more clearly discriminated on the basis of kinship, with higher quality among kin, and therefore potentially, a lower number/diversity of social partners). The same goes for the nature, diversity, and distribution of social interactions. However, as detailed in the calculation of the indices, I feel that the same system characteristics affect the measures of complexity investigated (e.g., see lines 309-314). Which, thus, overall, seems pretty circular to me.

Other comments:

Everywhere: Justify the text.

Abstract: Could authors make clearer in the abstract the reason why they decided to use Shannon's uncertainty formula (nicely explained lines 64-68).

Line 34: "sample" is "simple"?

Lines 3-4: Authors list is provided twice.

Line 35-37: This statement could be simplified to facilitate reading.

Lines 46-53. Authors refer to a number of "hypotheses" but list a number of questions.

Line 52-53: This last question seems a bit off-topic compare to the previous ones that more clearly refer to the complexity of social organizations.

Lines 65-66: “[...] at the essence of their complexity”?

Lines 115-116: Check the syntax

Line 126: Change [in (

Lines 162-163: “allowing them to explore functional abilities” It is not entirely clear what the authors mean here.

Line 247: Could authors provide the Hmax in the calculation examples of each index, and explain its calculation? This could help the reader to better understand the concept and definition of sample space.

Lines 258-261: the complexity of what? The social system/structure / network of relationships (as suggested lines 250-251, or 260)? At which level is the analysis made and following this, on which sample space the complexity indices will be calculated? Even later, based on the detailed calculations, it is not entirely clear what the authors aim to quantify with those three indices. Please clarify.

Decision letter (RSOS-200895.R0)

Dear Dr Thierry

The Editors assigned to your paper RSOS-200895 "Measuring complexity in organisms and organizations" have now received comments from reviewers and would like you to revise the paper in accordance with the reviewer comments and any comments from the Editors. Please note this decision does not guarantee eventual acceptance.

Please submit your revised manuscript and required files (see below) no later than 21 days from today's (ie 16-Nov-2020) date. Note: the ScholarOne system will 'lock' if submission of the revision is attempted 21 or more days after the deadline. If you do not think you will be able to meet this deadline please contact the editorial office immediately.

on behalf of Prof Kevin Padian (Subject Editor)
openscience@royalsociety.org

Associate Editor Comments to Author:

Thank you for your patience while we conducted the review of your paper. To secure the reviews received, we had to approach an unusual number of possible referees, no doubt in part a reflection of the difficulties many have experienced this year with COVID. In any case, based on the comments of reviewer 2 in particular, we're of the view that a major revision would be appropriate here. Bear in mind that multiple rounds of review are not generally possible, so please take every effort to fully respond to and edit your manuscript in response to the reviewer's commentary. Thanks again and we'll look forward to receiving your revision in due course.

Associate Editor: 2
Comments to the Author:
(There are no comments.)

Reviewer comments to Author:
Reviewer: 1

Comments to the Author(s)
Shannon entropy is a classic long-term concept, please at least including some empirical data from empirical systems to make the definition more solid and useful.

Reviewer: 2

Comments to the Author(s)
In this manuscript, authors introduce three new measures to quantify the complexity of systems. Built on information theory, their measures derive from the Shannon index measure of uncertainty, uncertainty being indeed, one of the most prominent characteristics of complex systems. Introducing three different indices to quantify complexity - diversity, flexibility, and combinability -, they also better acknowledge the multidimensional nature of complexity, considered as another notable property of complex systems. Doing so, they aim to provide a more comprehensible and comparable method for the study of systems' complexity, which was indeed, so far lacking. Doing so they point out the need for common operational and comparable measures of complexity, that could be easily applied to various systems. This certainly opens new possibilities and work avenues.

However, before recommending this manuscript for publication, I feel that substantial amendments/improvements regarding its structure and content are required.

First, in my view, the state of art regarding the definition of complexity (e.g., the different approaches/methods/measures that have been used to quantify it, and the different research fields that have address it) should be further developed. Doing so, I believe the work would benefit from a clearer positioning with respect to previous research (i.e., what has been done so far to quantify and compare complexity, and to which extend and why are we failing to do so), better highlighting its added value.

In the same line, I believe that the field of research mainly addressed in this work should be more explicit. In several sections (introduction, discussion) authors seem to address the field of animal social behavior, and in particular the study of social systems complexity (e.g., lines 48, 79). Following this, authors quantify the complexity of macaques' social system in their work example. And indeed, the study of social complexity across social systems is a hot topic. Such a contribution in the field would thus certainly be valuable.

However, the literature/framework in which this proposal is anchored, is never really explicitly mentioned, although this could substantially enrich the introduction and discussion (e.g., discussing the added value of their complexity measures with respect to the specific expectations and state of the art of the field), help the reader to identify the potential application of the work, and once more, better highlight its novelty.

E.g., introduction lines 64-68: when discussing the benefits of considering the outcomes of systems rather than their complexity per essence, it could be beneficial to the argumentation to provide a couple of examples applied to social systems to illustrate this particular point. This would also certainly ease the reading and contribute to illustrate potential applications of the proposed metrics and methodology.

It is sometimes difficult for the reader to picture at which analytic level the discussion (or calculations) is (are) made. For instance, the differences between elements, variables, types are not clear, and often confusing (e.g., lines 155-164). The use of those terms seems inter-changeable in some sections, while meaningful in others, although I'm sure their meaning might be clear to authors. I acknowledge the difficulty to find homogenous and logical terminologies to name such abstract concepts, but I think this is crucial to ensure a proper understanding of the methodology and its applications. A short section clearly defining what is meant by each 'technical' term, with a couple of examples, could be useful here.

At the end of the short introduction (line 107), could authors clearly say that the three following measures (diversity, flexibility, and combinability) are calculated using the exact same formula (Shannon's), but, applying it to different sample spaces?

Following this, the definition of the sample space appears crucial to this work. The whole method developed here thus rely on a proper understanding and definition of the sample space, for each of the three measures. However, I feel that this neither sufficiently highlighted nor explained with enough accuracy (first mentioned lines 133-136).

Another concern regards the subjectivity of the proposed metrics.

Again, if I understand correctly, the calculation of the three indices mostly relies on the specification of the sample space of each variable of interest. However, the choice of the elements/variables of interest and the definition of their sample space seems largely arbitrary to me. (Especially since the manuscript is not clearly positioned in a given field, and its associated guidelines, methodological challenges, or most prominently studied parameters not discussed). If the choice of elements/variables and the definition of their sample space is primarily determined

by the study system, I'm afraid it will mainly depend on the current knowledge of the system, and the type of data available (as pointed by authors lines 280-281, 404-408). Although corrected by Hmax (lines 123-128), I can imagine that the metrics would substantially vary depending on the choice of the subunits characterizing the system (e.g., when studying social systems: population, group, individual, to interaction level, as discussed lines 373-375); the nature and numbers of elements/variables considered; and the definition of their sample space.

My question is, therefore, how representative and reliable can those indices be?

For instance, in the working example, why using a single variable for the calculation of the Diversity index? And for this variable, which seems to be a demographic variable (this also has to be clarified), why are individuals chosen, as elements and differentiated only on the basis of their age and sex and no other characteristics e.g., sexual maturity, or considering the underlying genetic structure of the species which might be in particular relevant in species showing matriline - to which matriline cluster individuals belong -? Why not choosing a higher degree of refinement regarding age e.g., scaling age per year? Considering the full interaction between age and sex i.e., differentiating also male and female immatures? Similarly, for the flexibility index. Why quantifying behavioral variation and not relationships variation within the structure? And are the two kinds of social events considered (reconciliation and facial expression) sufficient to depict trustworthily behavioral variation across individuals' categories?

My problem here is that I feel that all those analytic choices can certainly be well justified. But they might all lead to different indices outcomes. This thus raises the question of comparability (between studies, systems). To ensure the reliable comparability of those indices, one would need to choose the exact same 'parameters' and sample space. Which, in my view represents the biggest challenge here, and therefore requires further discussion.

Could authors further discuss this, and possibly provide guidelines to help researchers how to properly choose the parameters of interest and their sample spaces, to consistently and comprehensibly investigate systems complexity?

Finally, no conclusions are drawn from the calculated indices in the working example, to answer the secondary objective of this work: "To evaluate whether complexity and tolerance correlate in macaque social organization, we calculated the complexity indices in two species of contrasting social relationships" Line 258-261. If authors do not aim to discuss their results but simply wish to detail their method, they should remove this statement. But I personally believe that concluding and discussing their results would improve the understanding of the measures and their potential applications and implications. But this remains to be done.

That being said, I am wondering whether the question isn't circular here (i.e., correlating the degree of tolerance to the calculated (social) complexity measures). Macaques' degree of tolerance is a by-product of the diversity, nature, and distribution of relationships and interactions in the groups. E.g., Compare to more tolerant systems, strong intolerance might be defined by more asymmetric and sparser relationships at the group levels, both for dominance relationships (i.e., clear dominants and subordinate status along with clear dyadic dominance relationships, and a larger proportion of dyads that likely never or very rarely directly 'encounter'), or affiliative relationships (i.e., relationships more clearly discriminated on the basis of kinship, with higher quality among kin, and therefore potentially, a lower number/diversity of social partners). The same goes for the nature, diversity, and distribution of social interactions.

However, as detailed in the calculation of the indices, I feel that the same system characteristics affect the measures of complexity investigated (e.g., see lines 309-314). Which, thus, overall, seems pretty circular to me.

Other comments:

Everywhere: Justify the text.

Abstract: Could authors make clearer in the abstract the reason why they decided to use Shannon's uncertainty formula (nicely explained lines 64-68).

Line 34: "sample" is "simple"?

Lines 3-4: Authors list is provided twice.

Line 35-37: This statement could be simplified to facilitate reading.

Lines 46-53. Authors refer to a number of "hypotheses" but list a number of questions.

Line 52-53: This last question seems a bit off-topic compare to the previous ones that more clearly refer to the complexity of social organizations.

Lines 65-66: "[...] at the essence of their complexity"?

Lines 115-116: Check the syntax

Line 126: Change [in (

Lines 162-163: "allowing them to explore functional abilities" It is not entirely clear what the authors mean here.

Line 247: Could authors provide the Hmax in the calculation examples of each index, and explain its calculation? This could help the reader to better understand the concept and definition of sample space.

Lines 258-261: the complexity of what? The social system/structure / network of relationships (as suggested lines 250-251, or 260)? At which level is the analysis made and following this, on which sample space the complexity indices will be calculated? Even later, based on the detailed calculations, it is not entirely clear what the authors aim to quantify with those three indices. Please clarify.

===PREPARING YOUR MANUSCRIPT===

While not essential, it will speed up the preparation of your manuscript proof if accepted if you format your references/bibliography in Vancouver style (please see

<https://royalsociety.org/journals/authors/author-guidelines/#formatting>). You should include DOIs for as many of the references as possible.

===PREPARING YOUR REVISION IN SCHOLARONE===

Author's Response to Decision Letter for (RSOS-200895.R0)

See Appendix A.

RSOS-200895.R1 (Revision)

Review form: Reviewer 2

Is the manuscript scientifically sound in its present form?

Yes

Are the interpretations and conclusions justified by the results?

Yes

Is the language acceptable?

Yes

Do you have any ethical concerns with this paper?

No

Have you any concerns about statistical analyses in this paper?

No

Recommendation?

Accept as is

Comments to the Author(s)

Authors have significantly improved their manuscript and perfectly answered all my questions and concerns.

In particular, by adding guidelines on how to select variables and sample size, they have greatly improved the understanding of their approach and I believe, provided guidelines that should be of great help to researchers wishing to apply this approach to their study system(s).

The different amendments made in the text have cleared up misunderstandings on authors' aims and greatly improved the clarity of the methods. Finally, they now better acknowledge the current limitations of their work, which for the moment mostly lies in the fact that the data needed to compute such indices are to date not available.

I, therefore, do not see any further points that could keep this manuscript from being published in its current form.

Decision letter (RSOS-200895.R1)

Dear Dr Thierry,

It is a pleasure to accept your manuscript entitled "Measuring complexity in organisms and organizations" in its current form for publication in Royal Society Open Science. The comments of the reviewer(s) who reviewed your manuscript are included at the foot of this letter.

===COVID-SPECIFIC TEXT -- WILL ONLY BE ADDED TO COVID-PAPERS BY THE EDITORIAL OFFICE===

COVID-19 rapid publication process:

We are taking steps to expedite the publication of research relevant to the pandemic. If you wish, you can opt to have your paper published as soon as it is ready, rather than waiting for it to be published the scheduled Wednesday.

This means your paper will not be included in the weekly media round-up which the Society sends to journalists ahead of publication. However, it will still appear in the COVID-19 Publishing Collection which journalists will be directed to each week (<https://royalsocietypublishing.org/topic/special-collections/novel-coronavirus-outbreak>).

If you wish to have your paper considered for immediate publication, or to discuss further, please notify openscience_proofs@royalsociety.org and press@royalsociety.org when you respond to this email.

===END OF COVID-SPECIFIC TEXT -- WILL BE REMOVED AS NECESSARY BY THE EDITORIAL OFFICE===

on behalf of Kevin Padian (Subject Editor)
openscience@royalsociety.org

Reviewer comments to Author:
Reviewer: 2

Comments to the Author(s)
Authors have significantly improved their manuscript and perfectly answered all my questions and concerns.

In particular, by adding guidelines on how to select variables and sample size, they have greatly improved the understanding of their approach and I believe, provided guidelines that should be of great help to researchers wishing to apply this approach to their study system(s). The different amendments made in the text have cleared up misunderstandings on authors' aims and greatly improved the clarity of the methods. Finally, they now better acknowledge the current limitations of their work, which for the moment mostly lies in the fact that the data needed to compute such indices are to date not available.

I, therefore, do not see any further points that could keep this manuscript from being published in its current form.

Appendix A

Dear Editor,

We would like to bring to your attention the revision of our manuscript entitled "Measuring complexity in organisms and organizations".

We thank reviewers for taking the time to read our manuscript and comment on it. This has allowed us to significantly improve our manuscript. You will find below the detail of the revision.

Best regards,
Bernard Thierry

Reviewer 1

Shannon entropy is a classic long-term concept, please at least including some empirical data from empirical systems to make the definition more solid and useful.

We give examples (l. 177-182, 232-236) and present a case of index calculation in section 2.5, where we explain at length how to calculate each index (l. 260-340) and how to combine them to obtain a complexity index (l. 343-353).

In the Electronic supplementary material, we use empirical data to show, step-by-step with numerical examples, how to calculate each of the diversity, flexibility and combinability indices.

Reviewer 2

In this manuscript, authors introduce three new measures to quantify the complexity of systems. Built on information theory, their measures derive from the Shannon index measure of uncertainty, uncertainty being indeed, one of the most prominent characteristics of complex systems. Introducing three different indices to quantify complexity - diversity, flexibility, and combinability -, they also better acknowledge the multidimensional nature of complexity, considered as another notable property of complex systems. Doing so, they aim to provide a more comprehensible and comparable method for the study of systems' complexity, which was indeed, so far lacking. Doing so they point out the need for common operational and comparable measures of complexity, that could be easily applied to various systems. This certainly opens new possibilities and work avenues.

However, before recommending this manuscript for publication, I feel that substantial amendments/improvements regarding its structure and content are required.

First, in my view, the state of art regarding the definition of complexity (e.g., the different approaches/methods/measures that have been used to quantify it, and the different research fields that have address it) should be further developed. Doing so, I believe the work would benefit from a clearer positioning with respect to previous research (i.e., what has been done so far to quantify and compare complexity, and to which extend and why are we failing to do so), better highlighting its added value.

After stating that we have to measure the three major dimensions of system complexity (i.e. diversity, flexibility and combinability), we now specify that "Much effort has been devoted to assessing the diversity of organisms and organizations by measuring the number and variety of their building blocks. However, diversity is only one component of complexity and we still lack indices capable of capturing the full extent of complexity." (l. 107-110).

We recall that richness and evenness have been to date the most frequently used indices to measure diversity (and thus complexity): "Richness is the number of possible outcomes, types, or categories of a variable. It is a popular measure of diversity/complexity as it is relatively easy

to count cell types in organisms [31], species in ecological communities [32], signals in animal communication [13,29], structures in languages [28], or cultural variants in human societies [6,33].” (l. 121-125)

In the same line, I believe that the field of research mainly addressed in this work should be more explicit. In several sections (introduction, discussion) authors seem to address the field of animal social behavior, and in particular the study of social systems complexity (e.g., lines 48, 79). Following this, authors quantify the complexity of macaques’ social system in their work example. And indeed, the study of social complexity across social systems is a hot topic. Such a contribution in the field would thus certainly be valuable.

However, the literature/framework in which this proposal is anchored, is never really explicitly mentioned, although this could substantially enrich the introduction and discussion (e.g., discussing the added value of their complexity measures with respect to the specific expectations and state of the art of the field), help the reader to identify the potential application of the work, and once more, better highlight its novelty.

E.g., introduction lines 64-68: when discussing the benefits of considering the outcomes of systems rather than their complexity per essence, it could be beneficial to the argumentation to provide a couple of examples applied to social systems to illustrate this particular point. This would also certainly ease the reading and contribute to illustrate potential applications of the proposed metrics and methodology.

We have taken the measure of the complexity of macaque social systems as an example of application, but we do not want to reduce the use of the complexity indices only to the study of animal social behavior, or even to the study of the social organization. In the introductory paragraph of the paper, we stress that the measure of complexity is an issue that concerns many fields in biological and social sciences (e.g. ecology, evolution, communication, animal societies, human societies, economics) (l. 44-54). To make our objectives more explicit, we have added the following sentence at the end of the Introduction:

“This should be of interest to all fields concerned with the complexity of living organisms and social organizations”. (l. 70-71)

We have also modified the last sentence of the paper with the same aim of clarification:

“Hopefully, the present proposal will encourage research programs that aim to measure the variables needed to compare the complexity levels of systems in both biological and social fields.” (l. 428-430)

It is sometimes difficult for the reader to picture at which analytic level the discussion (or calculations) is (are) made. For instance, the differences between elements, variables, types are not clear, and often confusing (e.g., lines 155-164). The use of those terms seems interchangeable in some sections, while meaningful in others, although I’m sure their meaning might be clear to authors. I acknowledge the difficulty to find homogenous and logical terminologies to name such abstract concepts, but I think this is crucial to ensure a proper understanding of the methodology and its applications. A short section clearly defining what is meant by each ‘technical’ term, with a couple of examples, could be useful here.

Please note that we already defined in lines 121-122 and 133-142, what are outcomes, types and variables. We also provide examples of our technical terms in lines 122-125 and 142-143.

However, it is true that we have used the words ‘types’ and ‘categories’ interchangeably since they are synonymous, which can be confusing for the reader. Therefore, we now avoid using ‘types’ and instead use ‘categories’. For the sake of clarity, we have also added that “When analyzing a system, we have first to specify what the elements of that system are, i.e. its basic units.” (l. 116-117)

At the end of the short introduction (line 107), could authors clearly say that the three following

measures (diversity, flexibility, and combinability) are calculated using the exact same formula (Shannon's), but, applying it to different sample spaces?

As requested, we now write:

“The measurement of complexity requires that all three dimensions are accounted for. As we develop below, the measures of diversity, flexibility and combinability will each be calculated using the Shannon's formula, applying it to different variables in corresponding sample spaces.” (l. 110-113)

Following this, the definition of the sample space appears crucial to this work. The whole method developed here thus rely on a proper understanding and definition of the sample space, for each of the three measures. However, I feel that this neither sufficiently highlighted nor explained with enough accuracy (first mentioned lines 133-136).

We defined the sample space of a variable of interest as the set of all possible outcomes (l. 141-142).

For the sake of clarity, we now specify that elements are the basic units of the system (116-117).

We also insist on the definitions at the beginning of each paragraph in the calculation example:

“Diversity concerns the characteristics of the elements that make up a system and by which these elements can be described.” (280-281); “In social systems, the elements are individuals, and their behavior may vary according to social situations (l. 294-295); “Since the elements of a social system are individuals, combinability concerns patterns of interactions and relationships between individuals” (l. 318-320).

Another concern regards the subjectivity of the proposed metrics.

Again, if I understand correctly, the calculation of the three indices mostly relies on the specification of the sample space of each variable of interest. However, the choice of the elements/variables of interest and the definition of their sample space seems largely arbitrary to me. (Especially since the manuscript is not clearly positioned in a given field, and its associated guidelines, methodological challenges, or most prominently studied parameters not discussed).

If the choice of elements/variables and the definition of their sample space is primarily determined by the study system, I'm afraid it will mainly depend on the current knowledge of the system, and the type of data available (as pointed by authors lines 280-281, 404-408). Although corrected by Hmax (lines 123-128), I can imagine that the metrics would substantially vary depending on the choice of the subunits characterizing the system (e.g., when studying social systems: population, group, individual, to interaction level, as discussed lines 373-375); the nature and numbers of elements/variables considered; and the definition of their sample space.

Subjectivity should not be confused with knowledge. The choice of variables is not arbitrary, but rather carefully considered in relation to current knowledge in the field. This comes back to the case where, in statistics, an experimenter tests the effect of explanatory factors on a response variable; considering factors and variables that make no sense for the system would lead to spurious results. Knowledge about the studied phenomenon allows us to avoid this, and this is also the case for our index. The choice of the sample space is far from being left to chance because it is based on uncertainty quantification with a focus on extreme cases. Moreover, the choice of sub-units is not left to chance either, and is directly derived from the basic elements that we have defined to study a given system.

To better explain how the selection of variables and sample spaces should be made, we have added a guidelines section in the Electronic supplementary material:

“GUIDELINES FOR SELECTING VARIABLES AND SAMPLE SPACES

We propose the following guidelines for the selection of variables, their dimension and the sample spaces to be used.

Selection of variables

Variables should be selected for their ability to measure the uncertainty of part of a system. This means ensuring that they are representative of the system based on what we know about it, and also that they are measurable in terms of uncertainty.

In the example of the social systems of macaques, social conflicts are characterized by different outcomes, with varying degrees of uncertainty in estimating the outcome of these interactions. We can therefore define a ‘social conflict’ variable based on the behavior of individuals involved in the aggression and the response to aggression.

Dimension of variables

Each variable should be assigned to a given dimension. The dimension of complexity that is measured by the variable – diversity, flexibility, combinability – depends on the level at which the elements of the system are recognized. For each variable, the following questions should be answered:

- Does the variable allow the uncertainty to be quantified in terms of the characteristics of the elements? If yes, the variable can be used in the calculation of diversity.
- Does the variable allow the uncertainty to be quantified in terms of element variations? If yes, the variable can be used in the calculation of flexibility.
- Does the variable allow the uncertainty to be quantified in terms of interaction between two or more elements? If yes, the variable can be used in the calculation of combinability.

In the example of macaque social systems, the variable ‘social conflict’ does not allow the uncertainty to be quantified either in terms of the characteristics of the elements – since the elements are the individuals – or in terms of the variation between the elements. However, it does allow uncertainty to be quantified as an interaction. This variable can therefore be used in the calculation of combinability.

Sample space

Once the variable is assigned to a given dimension, the associated sample space can be deduced. To define the sample space, the variable must be translated based on categories whose proportions can be calculated. It is enough to reason about extreme cases to validate the sample space. Two questions should be answered:

- Does the calculation carried out on the case corresponding to the highest level of uncertainty lead to the highest h_v value?
- Does the calculation carried out on the case corresponding to the lowest level of uncertainty lead to the lowest h_v value?

In the example of macaque social systems, there is more uncertainty about the outcome of social conflicts when opponents have similar rates of aggression. We can then consider the sample space ‘most frequent aggressor vs. less frequent aggressor’, using the proportion of aggression shown by each opponent. We may then consider extreme cases. The case where the proportion of aggression displayed by the most frequent aggressor is much higher than the proportion of aggression displayed by the less frequent aggressor, leading to a low value of h_v , which reflects a low level of uncertainty in the system. Indeed, if the most frequent aggressor displays a much higher proportion of aggression, we can reliably predict the outcome of a given conflict. On the contrary, if the two individuals display comparable rates of aggression, the calculation of h_v will lead to a higher value, which reflects the difficulty of predicting the outcome of a conflict, and thus a higher level of uncertainty in the system.”

My question is, therefore, how representative and reliable can those indices be?

For instance, in the working example, why using a single variable for the calculation of the Diversity index? And for this variable, which seems to be a demographic variable (this also has to be clarified), why are individuals chosen, as elements and differentiated only on the basis of their age and sex and no other characteristics e.g., sexual maturity, or considering the underlying genetic structure of the species which might be in particular relevant in species showing matriline – to which matriline cluster individuals belong -? Why not choosing a higher degree of refinement regarding age e.g., scaling age per year? Considering the full

interaction between age and sex i.e., differentiating also male and female immatures? Similarly, for the flexibility index. Why quantifying behavioral variation and not relationships variation within the structure? And are the two kinds of social events considered (reconciliation and facial expression) sufficient to depict trustworthily behavioral variation across individuals' categories?

My problem here is that I feel that all those analytic choices can certainly be well justified. But they might all lead to different indices outcomes. This thus raises the question of comparability (between studies, systems). To ensure the reliable comparability of those indices, one would need to choose the exact same 'parameters' and sample space. Which, in my view represents the biggest challenge here, and therefore requires further discussion.

Could authors further discuss this, and possibly provide guidelines to help researchers how to properly choose the parameters of interest and their sample spaces, to consistently and comprehensibly investigate systems complexity?

The reviewer's comment is of course relevant, but the indices we propose are new, and the data needed to calculate representative indices are therefore simply not available. We have added a sentence to acknowledge this, and thus justify the use of these variables:

"It should be noted that the data needed to calculate the indices are still scarce, so the variables selected were those for which the necessary information was available." (l. 275-277)

We do not claim that the variables chosen in our example are representative; they are only those variables for which the data were available. For greater clarity, we have added the following statement at the end of the text reporting the example:

"In addition, the results of the calculations depend on the variables chosen, so that the collection of data on a much larger number of variables would be necessary to obtain representative indices." (l. 352-353)

As previously mentioned, we now provide a guidelines section in the Electronic supplementary material that answers the reviewer's request.

Finally, no conclusions are drawn from the calculated indices in the working example, to answer the secondary objective of this work: "To evaluate whether complexity and tolerance correlate in macaque social organization, we calculated the complexity indices in two species of contrasting social relationships" Line 258-261. If authors do not aim to discuss their results but simply wish to detail their method, they should remove this statement. But I personally believe that concluding and discussing their results would improve the understanding of the measures and their potential applications and implications. But this remains to be done.

We are aware that the number of variables and the number of species are not sufficient to "demonstrate" that complexity and social tolerance significantly correlate. Our working example only aims to "illustrate" the calculation of indices (l. 261). We write: "It should be noted that we have used the example of macaque social systems to describe the calculation of the complexity index, but data from a higher number of species would be necessary to test whether the indices measured in different kinds of species are statistically different." (l. 348-351)

For the sake of clarity, we have corrected the sentence presenting our objectives as follows, "To illustrate how the correlation between complexity and tolerance can be evaluated in macaque social organization, we calculated the complexity indices in two species of contrasting social relationships." (l. 271-273)

That being said, I am wondering whether the question isn't circular here (i.e., correlating the degree of tolerance to the calculated (social) complexity measures). Macaques' degree of tolerance is a by-product of the diversity, nature, and distribution of relationships and interactions in the groups. E.g., Compare to more tolerant systems, strong intolerance might be defined by more asymmetric and sparser relationships at the group levels, both for dominance relationships (i.e., clear dominants and subordinate status along with clear dyadic dominance

relationships, and a larger proportion of dyads that likely never or very rarely directly ‘encounter’), or affiliative relationships (i.e., relationships more clearly discriminated on the basis of kinship, with higher quality among kin, and therefore potentially, a lower number/diversity of social partners). The same goes for the nature, diversity, and distribution of social interactions.

However, as detailed in the calculation of the indices, I feel that the same system characteristics affect the measures of complexity investigated (e.g., see lines 309-314). Which, thus, overall, seems pretty circular to me.

We believe that the circularity between tolerance and complexity arises from assumptions of the reviewer that we do not share. Our reasoning is not based on a simple relationship between social tolerance and diversity or complexity. We first examine systems and their elements in terms of information theory, and therefore uncertainty, and the probability of outcomes. We then calculate three indices (diversity, flexibility, combinability) that measures the uncertainty of systems, and that are general enough to be applicable to a wide variety of systems, including living organisms, social organizations and cultural systems. If a link is found between, for example, social tolerance and diversity, it stems from the unpredictability of the system. That should be now made clearer with the added guidelines that emphasize the importance of uncertainty in selecting variables and sample spaces.

Other comments:

Everywhere: Justify the text.

We have justified the text on the left.

Abstract: Could authors make clearer in the abstract the reason why they decided to use Shannon’s uncertainty formula (nicely explained lines 64-68).

We have added the following sentence : “While there is no consensus about the definition of complexity, it is widely accepted that the ability to produce uncertainty is the most prominent characteristic of complex systems.” (l. 25-26)

Line 34: “sample” is “simple”?

We now write “a calculation example” (l. 33).

Lines 3-4: Authors list is provided twice.

Fixed.

Line 35-37: This statement could be simplified to facilitate reading.

Fixed.

Lines 46-53. Authors refer to a number of “hypotheses” but list a number of questions.

Fixed.

Line 52-53: This last question seems a bit off-topic compare to the previous ones that more clearly refer to the complexity of social organizations.

The issues are not necessarily limited to social organizations, they may include economic development or the stability of ecological communities, for example (l. 47-52).

Lines 65-66: “[...] at the essence of their complexity”?

We have replaced “essence” by “nature” (l. 64)

Lines 115-116: Check the syntax

Fixed.

Line 126: Change [in (
Fixed.

Lines 162-163: “allowing them to explore functional abilities” It is not entirely clear what the authors mean here.

We now write “possible functional abilities” (l. 172).

Line 247: Could authors provide the H_{max} in the calculation examples of each index, and explain its calculation? This could help the reader to better understand the concept and definition of sample space.

We have added the calculation examples of the H_{max} for each index in the Supplement (Table 1), as requested.

Lines 258-261: the complexity of what? The social system/structure / network of relationships (as suggested lines 250-251, or 260)? At which level is the analysis made and following this, on which sample space the complexity indices will be calculated? Even later, based on the detailed calculations, it is not entirely clear what the authors aim to quantify with those three indices. Please clarify.

We now write “the complexity indices of the social system (l. 273)”.

As previously mentioned, we have added several sentences: “Diversity concerns the characteristics of the elements that make up a system and by which these elements can be described. Here, the elements are individuals, they can be described by their age and sex (l. 280-282); Flexibility is about how elements can vary. In social systems, the elements are individuals, and their behavior may vary according to social situations l. 294-295); Combinability refers to the connections between the elements of a system. Since the elements of a social system are individuals, combinability concerns patterns of interactions and relationships between individuals (l. 318-320).

The addition of a guidelines section in the Supplement now provides the additional explanations that are requested.